# Outcomes following early parenteral nutrition use in preterm neonates: protocol for an observational study

James Webbe, Nicholas Longford, Sabita Uthaya, Neena Modi, Chris Gale

Section of Neonatal Medicine, Imperial College London, London, UK

**Correspondence to**
Dr James Webbe;
j.webbe@imperial.ac.uk,
j.webbe@imperial.ac.uk

## ABSTRACT

**Introduction** Preterm babies are among the highest users of parenteral nutrition (PN) of any patient group, but there is wide variation in commencement, duration, and composition of PN and uncertainty around which groups will benefit from early introduction. Recent studies in critically unwell adults and children suggest that harms, specifically increased rates of nosocomial infection, outweigh the benefits of early administration of PN. In this study, we will describe early PN use in neonatal units in England, Wales and Scotland. We will also evaluate if this is associated with differences in important neonatal outcomes in neonates born between $30^{+0}$ and $32^{+6}$ weeks$^{+days}$ gestation.

**Methods and analysis** We will use routinely collected data from all neonatal units in England, Wales and Scotland, available in the National Neonatal Research Database (NNRD). We will describe clinical practice in relation to any use of PN during the first 7 postnatal days among neonates admitted to neonatal care between 1 January 2012 and 31 December 2017. We will compare outcomes in neonates born between $30^{+0}$ and $32^{+6}$ weeks$^{+days}$ gestation who did or did not receive PN in the first week after birth using a propensity score-matched approach. The primary outcome will be survival to discharge home. Secondary outcomes will include components of the neonatal core outcome set: outcomes identified as important by former patients, parents, clinicians and researchers.

**Ethics and dissemination** We have obtained UK National Research Ethics Committee approval for this study (Ref: 18/NI/0214). The results of this study will be presented at academic conferences; the UK charity Bliss will aid dissemination to former patients and parents.

**Trial registration number** NCT03767634

## Strengths and limitations of this study

► We will use routinely recorded data held in the National Neonatal Research Database (NNRD); these have been shown to be complete and of high quality.
► The NNRD covers the entire neonatal unit population of England, Wales and Scotland and thus provides power to explore associations between early parenteral nutrition use and outcomes.
► As a retrospective, observational study, results will be vulnerable to confounding.
► The NNRD contains a large number of variables that will assist in propensity score matching to form well-balanced groups, diminishing potential confounding.
► We will focus on core outcomes identified as important by former patients and parents, as well as clinicians and researchers.

## INTRODUCTION

Preterm birth abruptly ends the transplacental transfer of nutrients essential for fetal growth and development. Providing adequate nutrition is an essential component of neonatal care but very preterm infants often have difficulty in tolerating adequate volumes of milk immediately after birth.[1] To meet their nutritional needs, such infants are commonly given supplemental parenteral nutrition (PN). Preterm babies are among the highest PN users of all National Health Service (NHS) patients but clinical practice is variable both nationally and internationally: some neonatal units in high-income countries are reported to provide PN to around 70% of neonatal admissions,[2] while others report using no PN.[3] In the UK, national guidelines recommend administration of PN to all babies born below 30 weeks gestational age[4]; however, exact rates of PN use in UK neonatal units are not known. When considering the efficacy of PN use in neonates, although the impact of early PN on nitrogen balance is known,[5] the effects on survival, growth and neurodevelopment are less clear.[5 6] Despite widespread use, the impact of administration of PN on key outcomes has not been evaluated in randomised controlled neonatal trials powered for clinical end points.

PN carries well-established risks, of which the most serious and common is bloodstream infection.[7] There is a growing body of evidence that use of PN in critically unwell adults[8] and children[9] within the first 7 days of admission to an intensive care unit is associated with worse outcomes. Furthermore, a subgroup analysis of a paediatric intensive care unit

population limited to term neonates less than 28 days old showed an increase in nosocomial infection with early PN use.[10] These studies raise uncertainty over the balance of risks and benefits of PN administration in the early postnatal period. It is generally accepted that PN is likely to be beneficial in extremely preterm neonates where postnatal nutrient deficits are most severe and prolonged, but in moderately preterm neonates, the effect of PN use on neonatal survival or other key outcomes has never been conclusively demonstrated.[5 11 12] The largest randomised trials to date[13–15] have recruited fewer than 100 babies to each interventional arm and were therefore only powered to detect very large (10%) absolute risk differences in mortality or other outcomes. Neonates are vulnerable to unanticipated treatment effects across different organ systems[16]; therefore, it is important to show that PN does not adversely impact any important neonatal outcomes in addition to the nutrition outcomes commonly studied in PN trials. The scant evidence base to inform use is reflected in variation in timing of commencement, duration of use and composition of PN within and between countries.[17 18]

This protocol describes how we will approach the following research questions:

▶ What is the pattern of PN use in neonatal units in England, Scotland and Wales in the first 7 postnatal days?

▶ In neonates born between $30^{+0}$ and $32^{+6}$ weeks gestational age, is PN use in the first 7 postnatal days, compared with no use of PN, associated with altered survival to discharge home?

▶ For neonates born between $30^{+0}$ and $32^{+6}$ weeks gestational age, is use of PN in the first 7 postnatal days, compared with no use of PN, associated with different core neonatal outcomes?

## METHODS

### Study design

This study contains two projects that will use data held in the National Neonatal Research Database (NNRD). First, we will evaluate practice and then we will undertake a comparison of matched groups of moderately preterm babies.

### Data source

This study will use deidentified data held in the NNRD.[19] The NNRD holds data extracted from point-of-care electronic health records completed by health professionals during routine clinical care.[20] The Neonatal Data Set, a defined national data standard[21] comprising approximately 450 items, is extracted and transmitted to the Neonatal Data Analysis Unit at Imperial College London. The data set includes demographic items relating to mother and baby (eg, gestational age at birth, birth weight, maternal conditions), daily items (eg, feeding information, administration of PN), ad hoc items (eg, suspected infection, cranial ultrasound findings), discharge items (eg, diagnoses during admission, weight at discharge)

and 2-year follow-up data. It contains data about which hospital a baby is born in and which neonatal network (the managed clinical networks that provide neonatal care within a geographical are) each hospital is part of.[22] The NNRD holds data from all infants admitted to NHS neonatal units in England, Scotland and Wales (approximately 90 000 infants each year). In total, the NNRD contains data from approximately 1 million infants from 2008 to the present. Since 2012, all units in England and Wales have contributed data, and since 2015, all but one Scottish units have contributed, with complete coverage of Scottish neonatal units since 2018. The completeness and quality of data held in the NNRD has been shown to be high,[23] making it suitable for research.[24]

### Eligibility

For the description of practice, we will use data on all neonates born between 1 January 2012 and 31 December 2017 and admitted to a neonatal unit in England, Scotland and Wales.

For the matched comparative study, we will use data on all neonates born between $30^{+0}$ and $32^{+6}$ weeks gestational age between 1 January 2012 and 31 December 2017 and admitted to a neonatal unit in England, Scotland and Wales. Neonates with major congenital gastrointestinal malformations will be excluded as they cannot be fed enterally (online supplementary eTable 1). Neonates with life-limiting conditions[25] or congenital conditions requiring surgery in the neonatal period will be excluded as they will not receive standard neonatal nutritional care (online supplementary eTable 2). Both groups would bias results because they will fall predominantly within one arm of the study and will have systematically different outcomes from the wider population. Neonates with missing key background data (birth weight, sex or gestational age) or data for the primary outcome will also be excluded.

### Intervention

For the comparative study, the intervention of interest will be early PN. We will compare outcomes between two groups: *'PN'* and *'No PN'*. The *PN* group will comprise eligible infants who received PN at any point during the first 7 days after birth. Receipt of PN is defined as receiving any volume, of any type of PN (standardised or tailor-made), by any route (peripheral intravenous cannula or central venous catheter) for any duration. The *No PN* group will comprise eligible infants who did not receive any PN in the first 7 days after birth.

### Sample size

The descriptive study will include data from approximately 450 000 infants.

For the comparative study, we have calculated that 12 000 neonates are required in each group (*PN* and *No PN*) to have 90% power to detect (with two-sided significance of 5%) an absolute difference in survival to discharge of 1.3% between the groups. We calculated the

absolute difference expected using a baseline mortality rate of 3.4%[26] and an OR of 0.73 for early versus late PN suggested by previous research.[9]

Around 6000 neonates are born between $30^{+0}$ and $32^{+6}$ weeks postmenstrual age each year in England, Wales and Scotland. Thus, over the 5-year study period we will have 30 000 neonates in total. Pilot data from the NNRD suggest that 45% of this group of babies will receive PN so we anticipate having 13 500 neonates in the *PN* group and 16 500 in the *No PN* group.

## Outcomes

For the descriptive study, the primary outcome will be any use of PN in the first 7 postnatal days. For the comparative study, the primary outcome will be survival to discharge home; defined as recorded as alive at final neonatal unit discharge. Secondary outcomes for the comparative study will be the other components of the neonatal core outcomes set[27] :

► Late onset sepsis; defined in line with the Royal College of Paediatrics and Child Health National Neonatal Audit Programme (NNAP) definition 'pure growth of a pathogen from blood' or 'pure growth of a skin commensal' or a 'mixed growth' after the first 72 hours of life.[28]

► Necrotising enterocolitis (NEC); defined in line with the Royal College of Paediatrics and Child Health NNAP definition. NEC may be diagnosed at surgery, postmortem or on the basis of the following clinical and radiographic signs: at least one clinical feature from (1) bilious gastric aspirate or emesis, (2) abdominal distension, (3) occult or gross blood in stool (no fissure), and at least one radiographic feature from (1) pneumatosis, (2) hepatobiliary gas and (3) pneumoperitoneum.[28] As this definition was introduced in 2016 for cases where data have not been recorded, this alternative definition will be used; NEC is defined as a recorded diagnosis of NEC in an infant that received at least 5 consecutive days of antibiotics while kept nil by mouth.[29]

► Brain injury on imaging; defined as documented diagnosis of intraventricular haemorrhage (grade 3–4)[30] or cystic periventricular leucomalacia.

► Retinopathy of prematurity; defined as a record of any retinopathy of prematurity on routine screening in the National Neonatal Dataset 'retinopathy of prematurity ad-hoc form'.

► Bronchopulmonary dysplasia; defined in line with the Royal College of Paediatrics and Child Health NNAP definition of severe bronchopulmonary dysplasia 'receiving respiratory support at 36 weeks corrected gestational age'.[28]

► Need for surgical procedures; defined as any record of surgical procedure during the neonatal admission.

► Seizures; defined as any recorded diagnosis of seizures or seizure disorder.

► Growth; weight and head circumference, and SD score (SDS) of the weight and head circumference for postmenstrual age at discharge; weight velocity and head circumference velocity, and change in SDS of the weight and head circumference for postmenstrual age from birth to discharge.

► Blindness; defined as an answer of yes to the question 'Does this child have a visual impairment?' at 2 years of age.[28]

► Deafness; defined as an answer of yes to the question 'Does this child have a hearing impairment?' at 2 years of age.[28]

► Ability to walk; defined as an answer of yes to the question 'Is this child unable to walk without assistance?' at 2 years of age.[28]

The components of the core outcomes set quality of life, gross motor ability and cognitive ability will not be reported as relevant data are not captured in the NNRD.

## Data analysis plan

In the descriptive study, we will describe the characteristics of neonates that receive PN in the first 7 postnatal days. Infants will be grouped according to gestational age at birth (using WHO definitions),[31] birth weight (using the WHO classification),[32] by year of birth and by geographical region (at the level of neonatal network). Rates of PN use will be compared between different groups using the $\chi^2$ test.

In the comparative study, we will use propensity matching to minimise bias and confounding. We will ensure the two groups are as closely matched as possible except for the exposure of interest, administration of PN in the first week. Infants will be matched on gestational age at birth (in bands: $30^{+0}$ to $30^{+6}$, $31^{+0}$ to $31^{+6}$, $32^{+0}$ to $32^{+6}$), small for gestational age (treated as a dichotomous variable: <10th centile on the UK-WHO growth chart,[33] or ≥10th centile) and propensity score. The propensity scores will be divided into propensity groups by the method of splitting proposed by Imbens and Rubin, with appropriate trimming of babies with unusually high or low scores.[34] The propensity model will include maternal factors, infant factors at birth, infant factors occurring on the first day of birth (preceding the decision to administer PN) and organisational factors. A full list of background variables to be included in the propensity score can be found in online supplementary eText 1.

We will calculate absolute risk differences and ORs for the prespecified, dichotomous outcomes. All p values will be two-sided. We will use the Holm-Bonferroni method[35] when analysing secondary outcomes to avoid erroneous inferences due to the risk of false positives in multiple comparisons.

## Subgroup analyses

As part of the descriptive survey, we will compare PN use in infants of different gestational ages, different birth weights, in different geographical regions at neonatal network level,[22] and compare how PN use has changed over the 5-year period.

To better replicate the research undertaken in adult and paediatric randomised controlled trials,[8 9] which defined early PN use as being before 48 hours, we will undertake a sensitivity analysis comparing survival and all secondary outcomes in babies started on PN in the first 48 hours with those not receiving PN in the first 7 post-natal days using propensity score matching. The sample size will be reduced but this analysis will provide further data for future research.

We will undertake a sensitivity analysis of the comparative study to explore the possibility that an unobserved variable explains any effect size seen to minimise the risk of findings due to confounding. We will construct a dichotomous variable to stack the odds against the superior treatment option and then compare this to the observed background variables to explore whether it is plausible that such an unobserved variable exists.[36]

### Patient and public involvement
This project was planned and designed with input from two parents of preterm infants who had experience of PN. This project will measure outcomes that we identified as most important to over 400 stakeholders with experience of neonatal care, including former patients and parents.[27] Our study addresses three research priorities identified by the James Lind Alliance priority setting partnership for preterm birth,[37] namely prevention of infection, lung damage and NEC.

### Ethics and dissemination
This study will only use deidentified data already held within the NNRD. The NNRD is UK Research Ethics Committee (REC) approved (REC Reference: 16/LO/1093) and Confidentiality Advisory Group approved (ECC 8-05(f/2010)) and all data are stored on NHS servers. Parents can opt out of their baby's data being held within the NNRD. Study-specific REC approval and Health Research Authority and Health and Care Research Wales approval was obtained (18/NI/0214).

The results of this study will be presented at academic conferences and published in a peer-reviewed scientific journal. Bliss will aid dissemination in an appropriate form online and via social media.

## DISCUSSION
Uncertainty surrounds the use of PN in neonatal patients. Previous surveys of practice show that the use of PN in neonates is variable both nationally and internationally.[17 38 39] It is not known how PN is used in neonatal units in England, Scotland and Wales nor how this has changed over time,[40] and the gestational age at which the nutritional benefits of early PN outweigh the risks in moderately preterm babies is unknown. Clinical practice may vary due to the uncertainty surrounding the risks and benefits for individual babies; our study will give clinicians more evidence on which to base decisions about early use of PN.

The major limitation of any observational study is that the intervention is not randomly assigned and any differences in outcomes may be explained by confounding. We will address this issue using propensity score matching to generate two matched cohorts, analogous to the random allocation that would occur in a controlled trial. The NNRD contains comprehensive background data on both the infants and their mothers which will be included in the propensity score; these data have been demonstrated to be sufficiently robust for research purposes.[23 24] This means that we can ensure that any difference in outcomes seen is likely to be due to the exposure of interest rather than confounders. As propensity score matching only ensures that measured variables are balanced, we will undertake a sensitivity analysis to explore whether confounding due to an unmeasured variable is likely. Further strengths of this study are that data were entered by clinicians during routine care and so should not be subject to recall bias. The NNRD covers all neonatal units in England, Wales and Scotland, hence 'recruitment bias' due to incomplete population coverage is eliminated and findings will be generalisable to other populations.

In this work, we will describe rates of PN use in neonatal units across England, Scotland and Wales for the first time. We will also describe the rates of important neonatal outcomes in moderately preterm neonates who receive early PN. These data will allow future randomised controlled trials to calculate expected PN exposure rates so that recruitment can be planned and provide outcome rates to allow accurate sample size calculation. These prospective studies would ensure that all other elements of care are equivalent, controlling for any possible confounding factors and providing conclusive evidence of a causal link between PN use and outcomes. Identifying whether moderately preterm neonates benefit from early PN will guide practice and also inform future research. At present, there is uncertainty around optimal use of PN in preterm infants; this protocol describes a database study that is the first step in addressing this problem.

**Acknowledgements** We wish to thank Angela Richard-Löndt and Laura Noakes, parents of preterm infants, for their support in developing this research project and BLISS for their input and support developing the dissemination plan.

**Contributors** JW, CG and NM conceived this project. JW, CG and NL planned the statistical analyses. The first draft of the manuscript was written by JW and revised by NM; CG, SU, NL and NM edited and reviewed the manuscript. It was approved by JW, CG, NL, SU and NM.

**Funding** The funding for creating and maintaining the NNRD is from unrestricted funding awarded to NM. This includes costs involved in data transfer, storage, cleaning, merging, administration and regulatory approvals. The extraction of study data from the NNRD and analysis for this study is funded through a Mason Medical Research Fellowship awarded to JW. CG is funded by the United Kingdom Medical Research Council (MRC) through a Clinician Scientist Fellowship award (MR/N008405/1).

**Competing interests** JW has received support from Chiesi Pharmaceuticals to attend an educational conference and has received a research grant from Mason Medical Research Foundation. SU has received funding from the National Institute of Health Research, the Department of Health and Prolacta Life Sciences. SU has been on the Advisory Board of Fresenius Kabi and received honoraria and travel expenses for speaking at study days organised by Fresenius Kabi. SU is a member of the National Institute for Health and Care Excellence Parenteral Nutrition

Guideline Development Committee. NM is Director of the Neonatal Data Analysis Unit at Imperial College London. In the last five years, NM has served on the Board of Trustees of the Royal College of Paediatrics and Child Health, David Harvey Trust, Medical Women's Federation and Medact; and is a member of the Nestle Scientific Advisory Board. NM has received research grants from the British Heart Foundation, Medical Research Council, National Institute of Health Research, Westminster Research Fund, Collaboration for Leadership in Applied Health and Care Northwest London, Healthcare Quality Improvement Partnership, Bliss, Prolacta Life Sciences, Chiesi, Shire and HCA International; travel and accommodation expenses from, Nutricia, Prolacta, Nestle and Chiesi; honoraria from Ferring Pharmaceuticals and Alexion Pharmaceuticals for contributions to expert advisory boards, and Chiesi for contributing to a lecture programme. CG is part of an international team developing reporting guidance (a CONSORT extension) for clinical trials using cohorts and routinely collected health data. He has received support from Chiesi Pharmaceuticals to attend an educational conference; in the past 5 years he been investigator on received research grants from Medical Research Council, National Institute of Health Research, Canadian Institute of Health Research, Department of Health in England, Mason Medical Research Foundation, Westminster Medical School Research Trust and Chiesi Pharmaceuticals.

**Patient consent for publication** Not required.

**Provenance and peer review** Not commissioned; externally peer reviewed.

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
