## [Reviewer comments · BMJ Open]

ARTICLE DETAILS

TITLE (PROVISIONAL)	Outcomes following early parenteral nutrition use in preterm neonates: Protocol for an observational study
AUTHORS	Webbe, James; Longford, Nicholas; Uthaya, Sabita; Modi, Neena; Gale, Chris

VERSION 1 - REVIEW

REVIEWER	Walter Zingg University of Geneva Hospitals Switzerland
REVIEW RETURNED	11-Feb-2019

GENERAL COMMENTS	This is a valid study question with the aim to set the stage for a future randomized controlled trial investigating the benefit of parenteral nutrition (PN) in moderate preterm neonates with a gestational age (GA) between 30 0/7 and 32 6/7 weeks. Comments You compare no PN to “any” PN. It could be interesting to distinguish the PN-group further in receiving lipids+TPN or TPN only? The power calculation seems to be based on too optimistic data and a ratio between PN to no-PN that is not yet known. It is a good starting point but the final number of neonates to be included probably needs to be adjusted based on intermediate analysis. Minor comments Page 8; line 26: I am confused about the years: you mention England/Wales and Scotland submitting data from 2012 and 2015, respectively. The sentence before assumes that the database exists since 2008. Can you clarify? Page 9; line 45: mortality in your group of interest (30 0/7 – 32 6/7 GA) may be lower than 3.4%; this could inflate your required number of neonates. Page 9; line 56: here, you assume a proportion of PN/no PN of 45%/55% - where does this information come from given that learning about the proportion of neonates being given PN is one of your main research questions?
--

	Page 10; line 5: you may consider calculating also early and late mortality (7 days/ 28 days) to produce data that are comparable to other reports. Page 11; line 33: how are you getting these data, and how are you going to link them to the otherwise anonymous data from NNRD?
--	---

REVIEWER	Sascha Verbruggen, MD. PhD. ErasmusMC Sophia Children's Hospital, Rotterdam, The Netherlands
REVIEW RETURNED	11-Feb-2019

GENERAL COMMENTS	This manuscript describes a protocol of a retrospective observational study describing the use of Early PN in neonates born 30+0 - 32+6 weeks gestation. -The authors also claim to evaluate if early PN affects survival and other important outcomes. However, regarding the latter, the current study design does not allow to draw any causal conclusions. I think the authors best describe it as an association, in stead of causal wording such as 'affects' throughout the manuscript. -Furthermore, the authors base their study on results from adult and pediatric RCTs, which have a clear distinction between early PN (within 48 hours) versus late PN (>7 days); such distinction is not made in the current protocol. This means that the authors will also include PN initiation on day 6 as early PN? If so, I am not sure there will be a clinical/statistical difference between groups?? Furthermore, initiation of PN is now more a continuous variable if the authors want to address it this way. This requires different statistical methods. -Why have the authors chosen survival to discharge home as primary outcome? Instead of nosocomial infections of length of stay as was chosen in the pediatric RCT. -Will Enteral intake be corrected for or incorporated in the propensity score matching? Now, the risk, being an observational study, is that the sickest neonates are fed differently and this needs to be taken into account. -Considering the retrospective design, why only obtain data between 2012-2017?? Why not increase power with more data? -Why exclude neonates with congenital conditions / major surgery, who cannot be fed enterally? These were not excluded in the large RCTs in the adult and pediatric ICU trials? What is your hypothesis? I would not exclude these, maybe compare between groups? -eTable2: Q20-26; I do not understand / agree why these neonates would not get standard nutritional care, or should be excluded from the study? It appears that the authors are only looking for preterm neonates without any other condition, or risk of being critically ill? I do not think the authors should exclude these neonates with (for instance) a congenital heart defect. The results will not provide a clinical relevant answer if only the presence of premature birth makes a neonate eligible for inclusion in the analyses. I agree with the authors that by including them bias is a risk, because these neonates will be most sick, and thus have different nutritional management. However, what is the hypothesis or clinical relevant answer they want to pursue? By stratifying for these groups one might be able to study this very important group as well? -As explained above I see a problem with the definition of early and late PN: early being day 1-6, and late being > day 7...or NO PN at all
--

	-How are the authors managing neonates who have died or left the NICU <day 7, regarding their outcomes? Sensoring? -The authors have not described how the Enteral Nutrition is recorded and how they will use these important data in their analyses. The success rate of EN could be bias, and may be useful to use in Prop Score matching
--	---

REVIEWER	Ana Lucia Goulart Federal University of São Paulo São Paulo, SP Brazil
REVIEW RETURNED	25-Feb-2019

GENERAL COMMENTS	The subject of this manuscript is interesting and would bring contributions for neonatologists, nutritionists, pharmacists and others health professionals, in regard to administration of parenteral nutrition for preterm infants and possible risks associated with this therapy. However, I have some comments about the study, as follow:  - on pages 11 and 12 two different criteria for the classification of newborns - WHO and the UK-WHO growth curve, are presented. The authors could clarify which criteria will be used. - on pages 12 and 29, the authors defined low birth weight as any result entered in the BIRTH WEIGHT which is below the 10th centile on the UK-WHO growth chart. However, according to WHO low birth weight refers to newborns with birth weight below 2500g, regardless of gestational age. Newborns weighing below the 10th percentile of the growth chart should be characterized as small for gestational age - enteral nutrition and the use of breast milk or formula in the first seven days of life are not among the variables included in the study and may influence the results to be analyzed. Is it possible to include these variables in the study?
---

VERSION 1 – AUTHOR RESPONSE

Responses to reviewer comments

We are very grateful to the referees for their suggestions which we found useful.

REVIEWER COMMENTS:

Reviewer #1

Reviewer comment:

You compare no PN to “any” PN. It could be interesting to distinguish the PN-group further in receiving lipids+TPN or TPN only?

Response:

We are limited by the data that is available to us hence though interesting we are unable to separate the PN group further. We do not have details relating to whether babies received lipids or the specific type of PN.

Reviewer comment:

The power calculation seems to be based on too optimistic data and a ratio between PN to no-PN that is not yet known. It is a good starting point but the final number of neonates to be included probably needs to be adjusted based on intermediate analysis.

Response:

The power calculation is our best prediction for the number of babies we expect to have. As we do not at present know which UK babies are given PN it is difficult to predict how similar PN and No-PN babies will be and therefore how many babies will be lost in the matching process. However, our calculation suggests that even if 10% of babies are lost in the matching process both groups will still contain over the 12,000 babies required to power the study for mortality. The ratio of PN to No-PN babies is based on pilot data extracted from the NNRD. We have clarified this by expanding to the following: (Page 9, Line 56)

“Thus over the five year study period we will have 30,000 neonates in total. Pilot data from the NNRD suggests 45% of this group of babies will receive PN so we anticipate having 13,500 neonates in the PN group and 16,500 in the No PN group.”

Reviewer comment:

Page 8; line 26: I am confused about the years: you mention England/Wales and Scotland submitting data from 2012 and 2015, respectively. The sentence before assumes that the database exists since 2008. Can you clarify?

Response:

The database started in 2008 and has expanded since this point. It achieved population coverage for England and Wales in 2012, then for all but one neonatal unit in Scotland in 2015. We have rewritten this section for clarity (Page 8, line 31):

“In total the NNRD contains data from approximately one million infants from 2008 to the present. Since 2012 all units in England and Wales have contributed data and since 2015 all but one Scottish units have contributed.”

Reviewer comment:

Page 9; line 45: mortality in your group of interest (30 0/7 – 32 6/7 GA) may be lower than 3.4%; this could inflate your required number of neonates.

Response:

The mortality rate of 3.4% is based on data from the most recent MBRRACE-UK Perinatal Surveillance Report based on 2016 data, and is the best estimate we have for mortality in this population. If mortality rates do prove to be lower the required sample size would increase, but we see no reason to expect a large difference from the MBRRACE-UK data as these are so up-to-date.

Reviewer comment:

Page 9; line 56: here, you assume a proportion of PN/no PN of 45%/55% - where does this information come from given that learning about the proportion of neonates being given PN is one of your main research questions?

Response:

It is correct that establishing the proportion of neonates receiving PN is one of our research questions. Due to the large variation seen in international practice we undertook a small pilot data extraction of data from the NNRD to establish this proportion in a UK population. We have clarified this as follows (Page 9, Line 56) :

“Thus over the five year study period we will have 30,000 neonates in total. Pilot data from the NNRD suggests 45% of this group of babies will receive PN so we anticipate having 13,500 neonates in the PN group and 16,500 in the No PN group.”

Reviewer comment:

Page 10; line 5: you may consider calculating also early and late mortality (7 days/ 28 days) to produce data that are comparable to other reports.

Response:

Survival to discharge home is the most commonly reported outcome time point in large neonatal trials (Webbe et al. Inconsistent Outcome Reporting in Large Neonatal Trials: A Systematic Review. Paper to be presented at: Pediatric Academic Societies Meeting; April 27, 2019. Baltimore, MD.), and is meaningful to parents so we have chosen this measurement point to be consistent with previous research. Adding other time points increases the risk of multiple comparisons and complicates the interpretation of results.

Reviewer comment:

Page 11; line 33: how are you getting these data, and how are you going to link them to the otherwise anonymous data from NNRD?

Response:

Two year follow up data is also held within the NNRD and will be used for this study; therefore no external data linkage will be undertaken. To make this clearer we have added this to the description of the data source (Page 8, Line 12):

“The data set includes demographic items relating to mother and baby (e.g. gestational age at birth, birthweight, maternal conditions), daily items (e.g. feeding information, administration of PN), ad hoc items (e.g. suspected infection, cranial ultrasound findings), discharge items (e.g. diagnoses during admission, weight at discharge), and two year follow up data.”

Reviewer #2

Reviewer comment:

The authors also claim to evaluate if early PN affects survival and other important outcomes. However, regarding the latter, the current study design does not allow to draw any causal conclusions. I think the authors best describe it as an association, in stead of causal wording such as 'affects' throughout the manuscript.

Response:

This study is observational and so we can only aim to reduce the risks of confounding as much as possible. As we describe this work is a first step and if our findings are important further prospective research will be needed to explore a causal relationship. We have changed the two instances of the word 'affect' and added the following clarification in the discussion section:
(Page 3, Line 21)

“We will also evaluate if this is associated with differences in important neonatal outcomes in neonates born between 30⁺⁰ and 32⁺⁶ weeks^{+days} gestation.”

(Page 5, Line 12)

“The NNRD covers the entire neonatal unit population of England, Wales and Scotland and thus provides power to explore associations between early PN use and outcomes.”

(Page 15, Line 18)

“These prospective studies would ensure that all other elements of care are equivalent, controlling for any possible confounding factors and providing conclusive evidence of a causal link between PN use and outcomes. ”

Reviewer comment:

Furthermore, the authors base their study on results from adult and pediatric RCTs, which have a clear distinction between early PN (within 48 hours) versus late PN (>7 days); such distinction is not made in the current protocol. This means that the authors will also include PN initiation on day 6 as early PN? If so, I am not sure there will be a clinical/statistical difference between groups?? Furthermore, initiation of PN is now more a continuous variable if the authors want to address it this way. This requires different statistical methods.

Response:

Our study builds on the work in the RCT undertaken in ICU and PICU populations. As this work is not prospective we are not able to control the timing of PN administration in the same way that these studies did and had to choose thresholds to define early/late administration. We plan to treat PN as a dichotomous variable and so had to choose a threshold. Given the considerable uncertainties around PN use in the UK (e.g. we do not at present know what proportion of babies start PN on Day 1, 2, 3 etc.) this was not straightforward. We expect that most babies given PN will start within the first 48-72 hours of postnatal life and that only a minority of infants treated with PN will be started on Day 5-7. Placing the threshold at 48 hours of age would give insufficient time for the intervention to cause any separation in clinical course between the two groups, so in line with the previous RCTs the cut off of 7 days was used to define 'late PN'.

Splitting the population as described by the reviewer (only comparing babies started on Day 1/2 with babies started after Day 7) would increase the distinction between groups but would inevitably lead to a loss of sample size (as all babies commenced on PN on days 3-6 would not be eligible for either group). We feel that maintaining the sample size will allow adequate distinction between the two groups.

The reviewers suggested approach will provide useful information for the planning of future prospective studies and so we will undertake a subgroup analysis comparing babies started on PN in the first 48 hours with those not started on PN in the first seven days. We have added the following (Page 13, Line 1):

“To better replicate the research undertaken in adult and paediatric randomised controlled trials (8, 9) which defined early PN use as being before 48 hours we will undertake a sensitivity analysis comparing survival and all secondary outcomes in babies started on PN in the first 48 hours with those not receiving PN in the first seven postnatal days using propensity score matching. The sample size will be reduced but this analysis will provide further data for future research. ”

Reviewer comment:

Why have the authors chosen survival to discharge home as primary outcome? Instead of nosocomial infections of length of stay as was chosen in the pediatric RCT.

Response:

Survival was consistently ranked as the most important outcome to all stakeholders during the development of the neonatal core outcome set (Webbe et al. Core outcomes in neonatology: results of an international consensus process involving patients, parents, healthcare professionals and researchers. Paper presented at 7th Congress of the European Academy of Paediatric Societies; October 31, 2018. Paris, France.), and thus we think it is appropriate to consider it as the primary outcome.

Reviewer comment:

Will Enteral intake be corrected for or incorporated in the propensity score matching? Now, the risk, being an observational study, is that the sickest neonates are fed differently and this needs to be taken into account.

Response:

We agree with the reviewers that enteral intake is an important variable and one that we will correct for. As propensity score matching is only appropriate for variables that precede the intervention we will match on enteral feeding on Day 1 and 2 of postnatal life. We plan to treat initial enteral feeding as a categorical variable using the data held in the National Neonatal Research Database about type of milk received.

Our propensity score matching includes an extensive range of background covariates and so should ensure that the neonates in both groups are equally “sick” at the point where a decision about commencing early PN is made. If our groups do not match well (which will be obvious from the background variables of the two groups) then our results may be affected by many confounding variables.

To clarify this we have added the following (Page 30, Line 45)

Enteral feeding	Data extracted from DAILY CARE FLUIDS AND FEEDING ENTERAL FEED TYPE GIVEN on day 1 and 2  • Categorical: Only maternal milk feeding defined as any of 1 Breastfeeding, 2 Mothers fresh expressed breast milk, 3 Mothers frozen expressed breast milk on either day with no other code. • Categorical: Only donor milk feeding defined as 4 Donor expressed breast milk on either day with no other code. • Categorical: Only formula defined as only 6 Formula milk on either day with no other code. • Categorical: Not feeding defined as 9 - Not applicable (nil by mouth) on both days with no other code. • Categorical: Mixed feeding as any combination of codes not consistent with the above categories.
-----------------	---

Reviewer comment:

Considering the retrospective design, why only obtain data between 2012-2017?? Why not increase power with more data?

Response:

The data is only complete at a population level for England and Wales from 2012 and so we have chosen this time point as a cut off. While using earlier data would increase our sample size it could reduce the applicability of our analysis as we would not be using data from an entire population and there could be an overrepresentation of tertiary level units or surgical centres.

Reviewer comment:

Why exclude neonates with congenital conditions / major surgery, who cannot be fed enterally? These were not excluded in the large RCTs in the adult and pediatric ICU trials? What is your hypothesis? I would not exclude these, maybe compare between groups?

Response:

For our retrospective, observational methods to draw valid results we need comparable groups of patients who could reasonably have been treated with either intervention. The conditions we have excluded were life-limiting or necessitated surgery within the early newborn period and so are unlikely to be managed in the same way as preterm infants without these co-morbidities, they may fall predominantly in one group. In addition their outcomes are likely to be substantially different from unaffected babies. As an example preterm infants with congenital cardiac disease are reported to have mortality rates around 11%, considerably higher than the 3.4% we expect (Laas et al., Impact of preterm birth on infant mortality for newborns with congenital heart defects: The EPICARD population-based cohort study. BMC Pediatr. 2017;17(1):124.). This makes them a major source of bias: decisions about the intervention and eventual outcomes are both substantially affected by the underlying diagnosis.

Reviewer comment:

eTable2: Q20-26; I do not understand / agree why these neonates would not get standard nutritional care, or should be excluded from the study? It appears that the authors are only looking for preterm neonates without any other condition, or risk of being critically ill? I do not think the authors should exclude these neonates with (for instance) a congenital heart defect. The results will not provide a

clinical relevant answer if only the presence of premature birth makes a neonate eligible for inclusion in the analyses. I agree with the authors that by including them bias is a risk, because these neonates will be most sick, and thus have different nutritional management. However, what is the hypothesis or clinical relevant answer they want to pursue? By stratifying for these groups one might be able to study this very important group as well?

Response:

The exclusion criteria cover three groups: congenital gastrointestinal malformations, congenital life-limiting conditions and congenital conditions that will require surgery in the early neonatal period. We will include babies with many postnatal morbidities (both related to and independent of prematurity). Babies with these rare congenital conditions form a population that require different management and have different outcomes from the majority of 30-32⁺⁶ week babies cared for on a neonatal unit. They will receive different care and as discussed above will have starkly different outcomes from other babies. As the reviewer states including them risks introducing bias, and in this retrospective observational study we think that risk of introducing even a small systematic bias outweighs the benefit of including these groups. We want to provide information to help clinicians managing very preterm infants (most of whom will not have the rare congenital conditions we are excluding). Propensity score matching relies on a comparison between two similar populations, with the only difference being the intervention. The babies we plan to exclude (who are atypical babies, requiring atypical management and having atypical outcomes) could imbalance the groups. Even if they were included it is unlikely that it would be possible to match them (and thus they would effectively be informally excluded due to failure of matching). Ultimately, we anticipate that the total number of babies excluded due to these conditions will be very small. While evidence is needed for the nutritional management of babies with congenital cardiac disease and other conditions we think that this is beyond the scope of this project. It is an area we will consider looking into in the future.

Reviewer comment:

As explained above I see a problem with the definition of early and late PN: early being day 1-6, and late being > day 7...or NO PN at all

Response:

As described above we had to decide on certain cut offs to define the different groups. We also feel that the decision a clinician is making is whether, on day one of life, they should start PN or adopt a management strategy that does not include immediate PN. On Day 1 they will not know whether a baby will need PN later or not, and so we are comparing babies exposed to PN with those not exposed to PN as we feel this is the pragmatic question. This project is only the start and we hope our data will be able to guide future randomised controlled trials in which prospectively defined PN regimes will help to provide gold-standard evidence of optimal practice.

Reviewer comment:

How are the authors managing neonates who have died or left the NICU <day 7, regarding their outcomes? Sensoring?

Response:

Very few neonates of this gestation will have been discharged home by Day 7, and as we have population level coverage transfer between units is not an issue, so we do not anticipate this affecting our data.

Our primary outcome is survival to discharge home so neonates who die before Day 7 will not be excluded. We have excluded congenital life-limiting conditions so any death will be expected to associate with whether or not PN was started. As the primary outcome is survival the secondary

outcomes will be reported as prevalence rates within survivors (as any difference in mortality will have already been identified).

Reviewer comment:

The authors have not described how the Enteral Nutrition is recorded and how they will use these important data in their analyses. The success rate of EN could be bias, and may be useful to use in Prop Score matching

Response:

As discussed we agree that enteral feeding is important thus it will be treated as a categorical variable and included within the propensity score as described above. The data is recorded in the NNRD as a daily item detailing the type(s) of milk given to the baby in the previous 24 hours.

Reviewer #3

Reviewer comment:

on pages 11 and 12 two different criteria for the classification of newborns - WHO and the UK-WHO growth curve, are presented. The authors could clarify which criteria will be used.

- on pages 12 and 29, the authors defined low birth weight as any result entered in the BIRTH WEIGHT which is below the 10th centile on the UK-WHO growth chart. However, according to WHO low birth weight refers to newborns with birth weight below 2500g, regardless of gestational age. Newborns weighing below the 10th percentile of the growth chart should be characterized as small for gestational age

Response:

The two different criteria relate to the different sections of this study. The descriptive arm will include all neonates (preterm, term, normal weight, LBW, VLBW) and will use the WHO categories.

The comparative section of the study only includes very preterm infants and so the UK-WHO growth chart definition is more appropriate and the reviewer is absolutely correct that this population would be better described as small for gestational age. We have corrected this error as follows:

(Page 12, Line 16)

“Infants will be matched on gestational age at birth (in bands: 30+0 to 30+6, 31+0 to 31+6, 32+0 to 32+6), small for gestational age (treated as a dichotomous variable: <10th centile on the UK-WHO growth chart (33), or ≥10th centile) and propensity score.”

(Page 29, Line 37)

Small for gestational age	Small for gestational age group defined as Any result entered in the BIRTH WEIGHT which is below the 10 th centile on the UK-WHO growth chart Appropriate for gestational age group defined as All other babies
---

Reviewer comment:

- enteral nutrition and the use of breast milk or formula in the first seven days of life are not among the variables included in the study and may influence the results to be analyzed. Is it possible to include these variables in the study?

Response:

Enteral feeding on the first two days of life will be included in the propensity score as a background variable. The data we have is a daily summary of the different types of milk that a baby has received over the previous 24 hours and will be used in the propensity score. The categories will be separated

according to whether the babies receive exclusive maternal milk, exclusive donor milk, exclusive formula milk, a mixture of milks or no feeding at all.

To clarify this we have added the following (Page 30, Line 45)

Enteral feeding	Data extracted from DAILY CARE FLUIDS AND FEEDING ENTERAL FEED TYPE GIVEN on day 1 and 2  • Categorical: Only maternal milk feeding defined as any of 1 Breastfeeding, 2 Mothers fresh expressed breast milk, 3 Mothers frozen expressed breast milk on either day with no other code. • Categorical: Only donor milk feeding defined as 4 Donor expressed breast milk on either day with no other code. • Categorical: Only formula defined as only 6 Formula milk on either day with no other code. • Categorical: Not feeding defined as 9 - Not applicable (nil by mouth) on both days with no other code. • Categorical: Mixed feeding as any combination of codes not consistent with the above categories.
-----------------	---

VERSION 2 – REVIEW

REVIEWER	Verbruggen Erasmus MC Sophia children's Hospital
REVIEW RETURNED	08-Apr-2019

GENERAL COMMENTS	To my opinion the authors have corrected and change the manuscript regarding my most important comments.
--

REVIEWER	Ana Lucia Goulart Federal University of São Paulo São Paulo, SP, Brasil
REVIEW RETURNED	15-Apr-2019

GENERAL COMMENTS	I consider that my questions were appropriately answered and that the methodology presented for the research responds to the proposed objectives. The subject of this manuscript is interesting and would bring contributions to improve care for premature newborns.
---